# Frozen Elephant Trunk in Acute Aortic Syndrome: Retrospective Results from a Low-Volume Center

**DOI:** 10.3390/jcm14196697

**Published:** 2025-09-23

**Authors:** Andreas Voetsch, Roman Gottardi, Andreas Winkler, Domenic Meissl, Katja Gansterer, Rainald Seitelberger, Philipp Krombholz-Reindl

**Affiliations:** 1Department of Cardiac Surgery, Paracelsus Medical University, 5010 Salzburg, Austria; 2Institute of Innovative Cardiac Surgery, Karl Landsteiner University of Health Sciences, Division of Cardiac Surgery, University Hospital St. Poelten, 3100 St. Poelten, Austria; 3Department of Cardiovascular Surgery, Heart Centre Freiburg University, Faculty of Medicine, University of Freiburg, 79106 Freiburg, Germany; 4Department of Vascular and Endovascular Surgery, Paracelsus Medical University, 5010 Salzburg, Austria

**Keywords:** acute aortic dissection (AAD), frozen elephant trunk (FET), aortic surgery, aortic arch replacement

## Abstract

**Objective:** The role of the frozen elephant trunk technique in the treatment of acute aortic dissections is currently based on results from high-volume centers only. We investigated the patient selection process, intraoperative data, the evolution of surgical practice and outcomes from a low-volume center. **Methods:** A retrospective analysis was conducted on 202 acute aortic dissection (AAD) patients treated between October 2014 and December 2023. Patients were categorized into those receiving less invasive open aortic repair (group 1, n = 136) and those undergoing frozen elephant trunk procedures (FETs) (group 2, n = 66). Data on demographics, surgical procedures, and outcomes were analyzed. **Results:** Overall 30-day mortality was 16% (13% vs. 23%; *p* = 0.068). Rates of postoperative disabling stroke were similar (9% vs. 8%, *p* = 0.190). FET procedures required longer cardiopulmonary bypass (195 min vs. 234 min, *p* = 0.011), hypothermic circulatory arrest (26 min vs. 43 min, *p* < 0.001), and selective cerebral perfusion times (26 min vs. 47 min, *p* < 0.001). Follow-up indicated that 17% of FET patients received completion thoracic endovascular aortic repair (TEVAR) versus 4% in non-FET patients (*p* = 0.002), whereas no difference was seen in open surgical reintervention. Median follow-up at 33 months showed an overall mortality of 27%, with no significant difference between groups (23% in group 1 vs. 35% in group 2, *p* = 0.123). **Conclusions:** The FET technique is feasible in low-volume centers, yielding outcomes comparable to high-volume centers. FET proximalization and a liberal use of extra-anatomical left subclavian artery (LSA) grafts ease the learning curve. Completion treatments can be effectively conducted following FET implantation to further induce positive aortic remodelling.

## 1. Introduction

While high-volume centers have demonstrated favorable outcomes with the frozen elephant trunk (FET) technique, there is a lack of data from low-volume centers, which handle a smaller case load of acute aortic dissection (AAD) cases annually [1]. This raises questions about the generalizability of outcomes reported from high-volume centers [2]. The primary treatment goal is to achieve survival, and we need to balance the risk of extensive surgery against the potential long-term benefits of positive aortic remodelling. Almost three-quarters of patients present with dissection beyond the aortic arch [3,4]. Reported mortality ranges from 11% in Japanese series up to 18–25% in European series and the IRAD registry reports [5,6,7,8]. While the FET technique is a promising option, its efficacy and safety in the hands of low-volume centers remain uncertain, given the complexity and the required institutional expertise. While some patients may benefit from limited repair, others may need more extensive procedures. Current AHA guidelines provide a class IIb recommendation for antegrade stenting of the descending aorta if a dissection flap crosses the aortic arch to prevent late aortic events and treat malperfusion [9]. Several investigations demonstrated the effective promotion of positive downstream remodelling for the FET method [10,11,12]. European consensus documents recommend the FET technique in AAD to close proximal descending aortic entries, correct true lumen collapse, and treat malperfusion [13]. The most recent EACTS/STS Guidelines for diagnosing and treating acute and chronic syndromes of the aortic organ provide conceptual approaches for communications between lumina, especially if present on the outer curvature of the arch or in the proximal descending aorta [14]. While the FET technique currently represents a well-accepted treatment option in aneurysmal disease and chronic aortic dissection, divergent results are reported in the setting of acute aortic dissection [1,15,16,17].

This study aimed to evaluate the implementation of the FET technique for treating AAD in a low-volume center, performing 500–600 open heart procedures annually.

## 2. Patients and Methods

### 2.1. Study Population

This single-center study population comprises 202 consecutive acute aortic dissection patients who underwent open aortic repair in a low-volume center between October 2014 and December 2023. All patients who were referred and underwent surgical treatment for AAD at our center during the inclusion period were enrolled in this study. Inclusion criteria were AAD either type A, type non-A non-B, or type B with indication for surgical treatment.

In October 2014, the department started using the FET technique in AAD cases. AAD was defined as the onset of symptoms within 14 days [18]. All patients were operated as emergency cases in high/moderate hypothermic circulatory arrest. FET indications were concordant with the published EACTS/ESVS recommendations [11]. The study population was divided into group 1 (non-FET) and group 2 (FET). Two different types of prostheses were used in group 2: the branched Thoraflex hybrid graft (Vascutek, Terumo, Inchinnan, UK) and the E-vita open neo hybrid stent graft system in trifurcated design (Artivion, Hechingen, Germany). A 10 cm stent graft was used in 62 patients, while 2 received an 18 cm E-vita open neo prosthesis and another 2 a 15 cm Thoraflex hybrid graft for anatomical reasons. Individual risk estimation was assessed by EuroScore II and GERAADA score. The local ethics committee approved this retrospective study and waived the need for informed consent (Ethikkommission für das Bundesland Salzburg EK Nr: 1002/2021). Data analyses were performed in accordance with the Declaration of Helsinki and Good Clinical Practice.

### 2.2. Surgical Technique

The surgical technique for FET implantation involved median sternotomy and cardiopulmonary bypass with central arterial inflow (either the right axillary artery or direct aortic cannulation). High/moderate hypothermia at 26 °C rectal temperature and antegrade cerebral perfusion was utilized in all cases. Arch vessel management was determined by the individual surgeon on a case-by-case basis. The preference was for end-to-end anastomosis of the supra-aortic branches with or without extra-anatomical left subclavian artery bypass. Sizing of the FET prosthesis was based on the estimated maximal aortic diameter before dissection at the distal landing zone without oversizing as previously described [19,20].

Distal anastomosis proximal to Ishimaru zone 3 was employed in selected cases to facilitate the procedure. The surgical technique of proximalization and extra-anatomical LSA bypass graft has been previously described in detail by our group [21].

### 2.3. Data Collection

At our department all AAD patients are treated after index surgery in a specific aortic outpatient clinic according to contemporary recommendations [9,13]. Clinical data were retrospectively collected from the prospectively maintained institutional database and individual electronic health records. The clinical follow-up was completed in February 2024 by telephone interview of all patients alive.

### 2.4. Definition of Clinical Parameters

Proximalization of the FET prosthesis was defined according to the Ishimaru zones. Distal anastomosis in zones 0–2 was defined as proximalization. The severity of postoperative stroke was assessed using the modified Rankin Scale (mRS) and evaluated by consulting neurologists. Strokes causing no clinical symptoms (mRS 0), no significant disability (mRS 1), or slight disability (mRS 2) were classified as non-disabling strokes. Spinal cord injury was characterized by either permanent or transient paraplegia or paraparesis symptoms. Acute renal failure after surgery followed the RIFLE classification [22] for definition purposes. Distal stent graft-induced new entries (dSINEs) were defined as new intimal tear at the distal end of the FET in follow-up CT imaging.

### 2.5. Statistical Analysis

Statistical analysis was performed using IBM SPSS 29.0 for Mac (SPSS Software, IBM Corp., Armonk, NY, USA). All data are shown as n (%) or as median (first quartile/third quartile). Categorical variables were compared using the χ^2^ test with the calculation of exact *p*-values. Fisher’s exact test was used for the ‘expected’ cell size assumption (n ≤ 5). We used the Shapiro–Wilk test to verify normal distribution. The student’s *t*-test was used to compare the groups on normally distributed continuous variables, and the Mann–Whitney U-test was used to compare the groups when the continuous variable did not conform to a normal distribution. A *p*-value of <0.05 was defined as significant. Survival rates were estimated using the Kaplan–Meier method and compared between the groups using the log-rank test.

## 3. Results

### 3.1. Demographics, Risk Factors, and Clinical Status at Admission

The introduction of the FET technique and the proportion of FET procedures in the institutional AAD program over time are shown in Figure 1. Patient baseline characteristics and demographics are summarized in Table 1. In total, 202 patients aged 63 (54–72) years, of whom 70 were females, required emergent surgical repair of AAD. One third of the patients underwent FET implantation. Demographics and cardiovascular risk factors were comparable between groups. No difference in the frequency of preoperative disabling stroke (7% in group 1 vs. 11% in group 2; *p* = 0.405) was observed. Table 2 summarizes the underlying aortic pathology, including classification of dissection, location of most proximal entry tear, and clinical and radiographic signs of malperfusion. There were significant differences between groups in terms of classification of dissection (Stanford type A: 99% in group 1 vs. 77% in group 2; *p* < 0.001), most proximal entry tear site location (arch or descending: 21% in group 1 vs. 42% in group 2; *p* = 0.003), and freedom from radiographic signs of malperfusion (59% in group 1 vs. 36% in group 2; *p* = 0.004). Malperfusion syndrome, as the combination of radiographic signs and clinical symptoms of malperfusion, did not differ between groups (18% vs. 14%; *p* = 0.431).

### 3.2. Intraoperative Data

As shown in Table 3, FET procedures were routinely performed with central arterial inflow (either right axillary artery or direct aortic cannulation) at high/moderate hypothermia (26 °C). SCP was used in all FET cases, whereas 13% of non-FET cases were performed without SCP; *p* < 0.001. Unilateral or bilateral SCP was used at the discretion of the individual surgeon, based on preference without differences between groups. Concomitant root procedures were performed in 47% of cases without differences between groups (any root procedure: 52% in group 1 vs. 38% in group 2; *p* = 0.074). Additionally, concomitant CABG was performed more frequently in the non-FET group (17% in group 1 vs. 6% in group 2; *p* = 0.046). CPB, HCA, and SCP times were longer in the FET group (195 min vs. 234 min; *p* = 0.011, 26 min vs. 43 min; *p* < 0.001, 26 min vs. 47 min; *p* < 0.001), whereas CX time did not differ (120 min vs. 130 min; *p* = 0.509).

### 3.3. Evolution of FET Implantation Technique over Time

Extra-anatomical LSA bypass grafts and proximalization of the distal anastomosis according to Ishimaru zone were used in 24% and 51% of patients with a decline over time, as shown in Figure 2.

### 3.4. Clinical Outcomes

Outcome characteristics are summarized in Table 4. Overall, thirty-day mortality was 16%; mortality in the FET group was higher but did not reach statistical significance (13% vs. 23%; *p* = 0.068; non-FET vs. FET). Postoperative disabling stroke rate was 8% without differences between groups (7% vs. 12%; *p* = 0.190). We encountered one case of spinal cord injury in the FET group. Acute postoperative renal failure necessitating renal replacement therapy was observed in 16% of cases without differences between groups. Overall, 8% of patients underwent completion TEVAR to treat remaining or new disease-related pathologies during follow-up. FET patients were more frequently treated with TEVAR (4% vs. 17%; *p* = 0.002). The median time from index procedure to completion TEVAR was 35 (12–130) days. The most frequent indication for completion TEVAR during follow-up was downstream true lumen compression in 38%, followed by aortic dilatation in 25% and coverage of new aortic lesions or large intimal tears in 19%, whereas dSINE was the indication 13%. The incidence of dSINE after FET was 3% during a median follow-up of 27 (1–58) months. All cases were completed without morbidity or mortality in a percutaneous endovascular approach. Aorta-related reoperation was performed in 6% of cases in both groups, *p* = 1.000.

Follow-up was completed in February 2024 and available in all patients. At a median follow-up time of 33 (1–69) months, mortality is 27% without significant differences between groups (23% in group 1 vs. 35% in group 2; *p* = 0.123), as shown in Figure 3.

## 4. Discussion

One third of AAD patients were treated with frozen elephant trunks in our low-volume center since the implementation of the FET technique in 2014. Baseline characteristics were well balanced between groups: whereas aortic characteristics according to TEM classification differed in terms of most proximal entry tears’ site location and radiographic signs of malperfusion, no differences were observed in clinical relevant malperfusion syndrome and supra-aortic vessel involvement [23]. Morphologic characteristics of dissection determined the treatment approach according to recent guidelines [14]. Our FET implantation technique followed the principles of central cannulation (either direct aortic or right axillary artery), high moderate hypothermia, and antegrade cerebral perfusion in all cases. The zone of distal anastomosis varied over time with a liberal use of proximalization and extra-anatomical LSA bypass grafts in the first cases. Details of our operative technique have been previously described [21]. The extent of repair is a constant point of discussion, determined not only by guideline recommendations but also by the local experience of the center. The primary goal is to save the patient’s life, but the long-term prognosis must not be disregarded. The increased technical complexity and the resulting potentially increased perioperative mortality of total arch replacement have repeatedly led to criticism of the FET method in ATAAD and a two-step approach towards the aortic arch may be justified [24]. Nevertheless, high-volume aortic centers frequently report equivalent perioperative mortality without increased stroke and paraplegia rates in non-randomized data sets and emphasize long-term benefits of extensive arch surgery. Meta-analysis of available data and registry data confirmed these findings [25,26,27,28]. Residual downstream dissection is associated with aneurysmal formation and aortic reintervention in a relevant proportion of patients [29]. Positive aortic remodelling induced with the FET technique may reduce future aortic interventions, and positive remodelling was demonstrated at the level of the descending aorta, whereas positive remodelling of the abdominal aorta is less frequently seen [30]. Recently reported data demonstrate improved intermediate survival after FET implantation compared to less extensive repair in ATAAD when performed by experienced professionals [1]. Data on outcome after extensive arch replacement in ATAAD performed in low-volume centers are currently not available. Whether extended arch surgery influences the treatment decision at the level of the aortic root needs to be discussed. For experienced aortic surgeons, the combination of aortic root and arch repair does not negatively impact outcome [31]. Neurological outcome in aortic surgery is greatly influenced by HCA and SCP times [32]. Although HCA and SCP times are prolonged in FET cases, we have not observed impaired neurological outcomes in this series. This may be attributable to the concept of FET proximalization and extra-anatomical LSA grafts, which was previously shown to ease the procedure at the beginning of the learning curve. Thereby, HCA and SCP times could be kept within an acceptable range of 43 (34–43) and 47 (37–62) minutes. These short HCA times made it possible to avoid a decline in root repairs. This is also reflected throughout the literature, with improved results regarding mortality, as well as intra- and postoperative complications when performing zone 2 FET vs. zone 3 FET [33]. Residual aortic dissection after initial repair may progress into late aortic dilatation and contribute as a risk factor for late mortality. Dilatation can occur as late as a decade after initial repair [34]. Newer techniques such as the Ascyrus Medical Dissection Stent (AMDS; Ascyrus Medical, Boca Raton, FL, USA) comprising a limited ascending aortic replacement with uncovered stenting of the arch and proximal descending aorta are still under investigation. Potential benefits are the less invasive implantation with limited circulatory arrest times comparable to hemiarch reconstruction, a technically easy anastomosis in zone 0 proximal to the innominate artery, and the potential benefit of re-expanding the true lumen, promoting aortic arch remodelling, reducing the risk of distal malperfusion, and reducing the risk of distal anastomotic new-entry tear (DANE). A recent meta-analysis on this topic compared the outcomes of 319 patients treated with the AMDS stent to 4129 patients treated with a FET [35]. In this meta-analysis the FET technique was associated with lower mortality and stronger long-term evidence. But not only residual dissection may play a role in treatment progression. The FET itself can cause new entry tears, and reinterventions after FET are frequent [35]. We performed completion TEVAR in 17% of FET patients during follow-up to treat residual or new pathology without mortality or morbidity. The FET served as a perfect landing zone in all procedures. The high rate of completion TEVAR in the FET group shows the low threshold for complementary interventions in contrast to conservative pre-treated patients. In contrast, open surgical treatment was rarely used. Longer follow-up will be needed to assess the impact on aorta-related outcomes and its impact on long-term mortality. Although the difference was not statistically significant, the 30-day mortality rate was higher in the frozen elephant trunk (FET) group compared to the non-FET group (23% vs. 13%). Notably, the GERAADA score, specifically designed to estimate the mortality risk for patients undergoing surgery for acute aortic dissection, was elevated in the FET group (20% vs. 16%), indicating a more critical condition among these patients. This disparity should be considered when interpreting this study’s findings. Conversely, the EuroScore II, which is not tailored to predict outcomes in acute aortic dissection repair, was lower in the FET group (7.4% vs. 4.7%). The lower EuroScore II may be due to the younger median age and the lower percentage of female patients in the FET cohort.

### Limitations and Strengths

This study has, like all retrospective, single-center studies, its inherent limitations. It should be emphasized that this study was not designed to compare outcomes between FET and less extensive aortic repair groups, and any observed differences should be interpreted in the context of differing risk profiles and procedural complexity. Both groups were well balanced in their baseline characteristics except signs of malperfusion and arch entry tears, which account for most FET indications. As a result, we opted not to perform propensity score matching on the cohorts. Additionally, despite being a low-volume center, all patients were treated by a specially trained aortic team. Therefore, a direct transfer of the results to other low-volume centers is not possible without restrictions.

## 5. Conclusions

The frozen elephant trunk technique, in acute aortic dissection, can safely be utilized in low-volume centers with results comparable to high-volume centers. Proximalization of the FET and a liberal use of extra-anatomical LSA bypass grafts may be a key technique to ease the learning curve and to build confidence in extensive AAD repair. Despite prolonged HCA and SCP times compared to less extensive arch surgery, we did not encounter a negative impact on neurological outcomes or mortality. Furthermore, no decline in concomitant root procedures was seen after implementing total arch repair in clinical practice. During follow-up, downstream treatment with completion TEVAR can easily be performed after FET implantation. We believe the FET method can be offered in low-volume centers with excellent results if performed by a specially trained aortic team.

**Key question.** Total arch replacement in the frozen elephant trunk technique (FET) promotes positive downstream aortic remodelling in the treatment of acute aortic dissection. Does the outcome justify extensive arch surgery in low-volume centers?

**Key findings.** Low-volume centers can achieve reasonable results with the frozen elephant trunk method in acute aortic dissection. The FET technique does not increase perioperative morbidity or mortality when used according to recommendations. Technical details such as the generous use of an extra-anatomical bypass graft to the left subclavian artery and proximalization of the distal anastomosis to zone 2 or even zone 1 ease the procedure, especially at the beginning of the learning curve.

**Take-home message.** The FET method should not be reserved for high-volume centers treating acute aortic dissections. If indicated, the FET technique can be used to induce positive downstream aortic remodelling at acceptable operative risk and provides a suitable landing zone for future endovascular treatment in AAD patients.

## Figures and Tables

**Figure 1 jcm-14-06697-f001:**
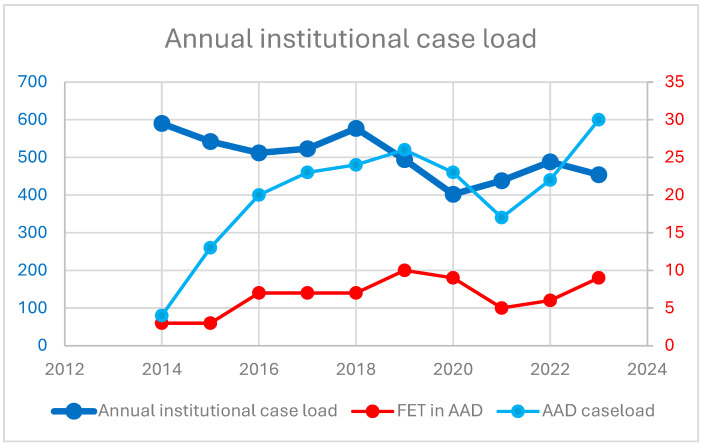
Annual institutional case load. Y-axis refers to number of cases performed annually, displaying the institutional case load, AAD cases, and FET procedures in AAD.

**Figure 2 jcm-14-06697-f002:**
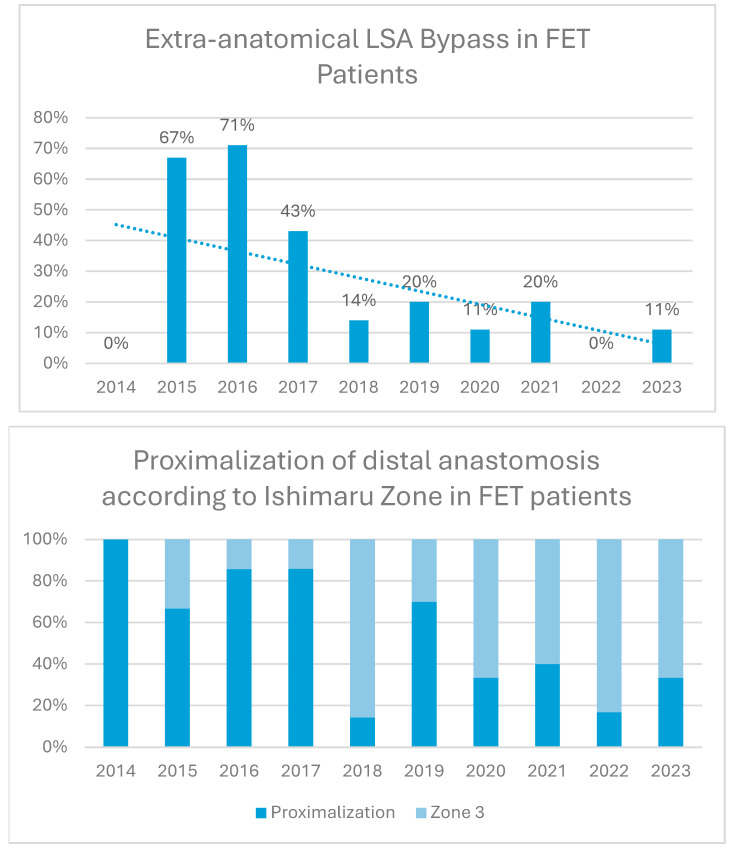
Proximalization and left subclavian artery (LSA) bypass trends over time. Proximalization refers to distal aortic anastomosis proximal to the LSA.

**Figure 3 jcm-14-06697-f003:**
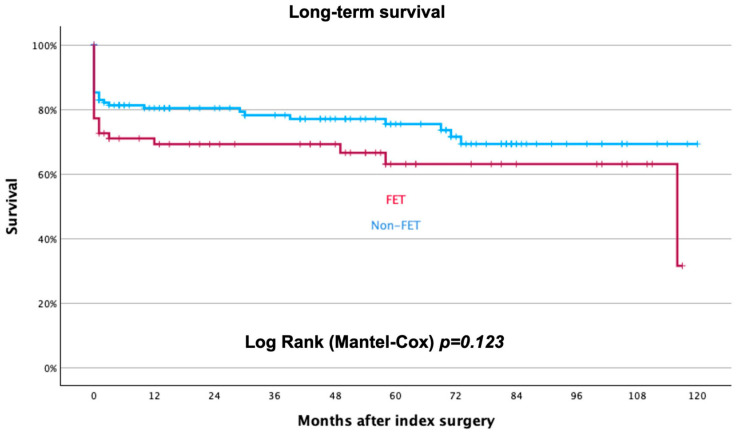
Long-term survival rate after index treatment for acute aortic dissection. Blue curve represents non-FET patients, red curve FET patients.

**Table 1 jcm-14-06697-t001:** Demographics.

	All	Non-FET	FET	*p*-Value
Patients n (%)	202	136 (67)	66 (33)	
Age [years]	63 (54–72)	65 (54–72)	60 (53–70)	0.213
Female sex	70 (35)	53 (39)	17 (26)	0.083
Delay diagnosis—surgery [h	2 (1–4)	2 (1–4)	2 (1–4)	0.131
Preoperative renal failure	25 (12)	18 (13)	7 (11)	0.656
BMI [kg/m^2^]	26 (24–30)	26 (24–29)	26 (24–30)	0.460
Preoperative coronary artery disease	25 (12)	18 (13)	7 (11)	0.656
Preoperative stroke	16 (8)	9 (7)	7 (11)	0.405
Preoperative hemiparesis	11 (5)	9 (7)	2 (3)	0.509
Clinical symptoms of malperfusion	34 (17)	25 (18)	9 (14)	0.431
Intubation at referral	10 (5)	8 (6)	2 (3)	0.503
Inotropic support	9 (5)	8 (6)	1 (2)	0.276
Aortic regurgitation ≥ moderate	47 (23)	33 (24)	14 (21)	0.413
EuroScore II	6.1 (3.6–12.5)	7.4 (4.3–7.4)	4.7 (3.3–8.4)	0.070
GERAADA score	18.2 (11.2–26.6)	16.4 (10.7–26.3)	20.1 (13.0–27.0)	0.311

Data are presented as number (%) or median (first quartile/third quartile). BMI: body mass index.

**Table 2 jcm-14-06697-t002:** Aortic characteristics.

	All (n = 202)	Non-FET (n = 136)	FET (n = 66)	*p*-Value
Classification				**<0.001**
Stanford type A	185 (92)	134 (99)	51 (77)	
Stanford type B	5 (3)	0 (0)	5 (8)	
Type non-A non-B	12 (6)	2 (2)	10 (15)	
Most proximal entry tear location				**0.021**
None	22 (11)	15 (11)	7 (11)	
Root	50 (25)	38 (28)	12 (18)	
Ascending	73 (36)	54 (40)	19 (29)	
Arch	46 (23)	25 (18)	21 (32)	
Descending	11 (5)	4 (3)	7 (11)	
Supra-aortic vessel involvement	97 (48)	66 (49)	31 (47)	0.881
Radiographic signs of malperfusion				
None	104 (52)	80 (59)	24 (36)	**0.022**
Coronary	9 (5)	6 (4)	3 (5)	
Visceral	32 (16)	18 (13)	14 (21)	
Peripheral	15 (7)	9 (7)	6 (9)	
Unknown	18 (9)	7 (5)	11 (17)	
Multiple	24 (12)	16 (12)	8 (12)	
Malperfusion syndrome	25 (18)	9 (14)	34 (17)	0.431

Data are presented as numbers (%). Values in boldface indicate statistical significance.

**Table 3 jcm-14-06697-t003:** Intraoperative data.

	All (n = 202)	Non-FET (n = 136)	FET (n = 66)	*p*-Value
Arterial cannulation				
Aortic	92 (46)	60 (44)	32 (49)	0.283
Axillary	100 (50)	67 (49)	33 (50)	
Femoral	10 (5)	9 (7)	1 (2)	
Lowest body temperature [°C]	26 (25–27)	26 (24–28)	26 (25–27)	0.991
Cerebral perfusion strategy				
None	25 (10)	25 (13)	0 (0)	**0.001**
Unilateral	104 (42)	78 (40)	26 (47)	
Bilateral	119 (48)	90 (47)	29 (53)	
Operative times [min]				
CPB	211 (170–265)	195 (158–261)	234 (189–234)	**0.011**
HCA	32 (23–44)	26 (21–35)	43 (34–43)	**<0.001**
SCP	33 (21–50)	26 (19–38)	47 (37–62)	**<0.001**
CX	124 (88–160)	120 (85–165)	130 (94–158)	0.509
Arch procedure				
None	19 (9)	19 (14)	0 (0)	**<0.001**
Hemiarch	93 (46)	93 (68)	0 (0)	
Conv. arch replacement	24 (12)	24 (18)	0 (0)	
FET	66 (33)	0 (0)	66 (100)	
Root procedure				
Supracoronary replacement	107 (53)	66 (49)	41 (62)	0.186
Selective Sinus replacement	5 (3)	2 (2)	3 (5)	
Bentall	68 (34)	52 (38)	16 (24)	
David	10 (5)	7 (7)	3 (5)	
Yacoub + Ring	12 (6)	9 (7)	3 (5)	
FET implant technique				
Zone 0			0 (0)	**<0.001**
Zone 1			3 (5)	
Zone 2			31 (47)	
Zone 3			32 (49)	
Extra-anatomical LSA bypass	24 (12)	8 (6)	16 (24)	**<0.001**
Concomitant cardiac procedures				
CABG	27 (13)	23 (17)	4 (6)	**0.046**
Mitral valve repair/replacement	11 (6)	5 (4)	6 (9)	0.183
Tricuspid valve repair	14 (7)	9 (7)	5 (8)	0.775

Data are presented as numbers (%) or median (first quartile/third quartile). Values in boldface indicate statistical significance. CPB: cardiopulmonary bypass; HCA: hypothermic circulatory arrest; SCP: selective cerebral perfusion; CX: cross-clamp; LSA: left subclavian artery; CABG: coronary artery bypass grafting.

**Table 4 jcm-14-06697-t004:** Clinical and aortic outcomes.

	All (n = 202)	Non-FET (n = 136)	FET (n = 66)	*p*-Value
Disabling stroke	17 (8)	9 (7)	8 (12)	0.190
Spinal cord injury	1 (1)	0 (0)	1 (2)	0.327
Postoperative renal replacement therapy	33 (16)	20 (15)	13 (18)	0.418
Ventilation time [hours]	29 (12–94)	28 (12–101)	31 (12–84)	0.618
Secondary distal extension (TEVAR)	16 (8)	5 (4)	11 (17)	**0.002**
Aorta-related reoperation during FU	6 (6)	4 (6)	2 (6)	1.000
30-day mortality	32 (16)	17 (13)	15 (23)	0.068
Long-term mortality *	54 (27)	31 (23)	23 (35)	0.123

Data are presented as number (%) or median (first quartile/third quartile). Values in boldface indicate statistical significance. TEVAR: thoracic endovascular aortic repair. * Mortality during follow-up was compared by log-rank test.

## Data Availability

The raw data supporting the conclusions of this article will be made available by the authors on request.

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
