# Peer review of "Frozen Elephant Trunk in Acute Aortic Syndrome: Retrospective Results from a Low-Volume Center"

_jcm, 2025, doi:10.3390/jcm14196697_

Round 1

Reviewer 1 Report

Comments and Suggestions for Authors

I was pleased to review this manuscript entitled "Frozen elephant trunk in acute aortic syndrome – results from a low-volume center".  This article investigates the implementation of the FET technique for treating acute aortic dissection in a low-volume center, performing 500-600 open-heart procedures annually. The authors concluded that the FET technique is feasible in low-volume centers, yielding outcomes comparable to high-volume centers. The manuscript is well-written, and the topic is interesting.

1) Please explain all abbreviations in the abstract section, such as FET, AAD, etc, as well as in the main text.

2) I would suggest adding to the title of this manuscript that this is a retrospective analysis.

3) I would suggest adding at least 2 more keywords

4) In the materials and methods section, make more clear the inclusive and exclusive criteria of the chosen study population

5) In this study, 202 were included. What was the minimum number of patients required to have safe results?

6) In the table that included demographic information of patients, could you also add the American Society of Anesthesiologists Physical Status (ASA-PS) classification score?

Author Response

Response to Reviewer 1 Comments

  1. Summary

1)         Please explain all abbreviations in the abstract section, such as FET, AAD, etc, as well as in the main text.

Thank you for your comment. We have added explanations for abbreviations throughout the manuscript to improve clarity. To maintain clarity, explanations for commonly known societies and standardized clinical scores (such as RIFLE) are not included in the main text but are listed in the glossary, as these terms are generally familiar to the intended audience.

2)         I would suggest adding to the title of this manuscript that this is a retrospective analysis.

Thank you for the suggestions. The term retrospective is added to the title.

3)         I would suggest adding at least 2 more keywords

Thank you for your suggestions, we added the keywords “aortic surgery” and “aortic arch replacement”.

4)         In the materials and methods section, make more clear the inclusive and exclusive criteria of the chosen study population

Thank you for your comment, we added a sentence on inclusion criteria to this section. Inclusion criteria used: all referred acute aortic dissection cases who underwent surgical treatment during the inclusion period.

5)         In this study, 202 were included. What was the minimum number of patients required to have safe results?

Thank you for your comment. As this was a retrospective observational study, no formal sample size calculation was performed prior to data collection. However, the final sample of 202 patients including 66 FET patients represents a substantial cohort for this condition, which is relatively uncommon in low-volume centers, and allows for meaningful descriptive and comparative analyses. We acknowledge this as a limitation and have added a note in the discussion accordingly.

6)         In the table that included demographic information of patients, could you also add the American Society of Anesthesiologists Physical Status (ASA-PS) classification score?

We appreciate your suggestion. However, because acute aortic dissection is an acute, life-threatening emergency, all patients in our cohort presented in a critical condition and were classified as ASA Physical Status 4 (or higher in some cases). As such, the ASA score does not provide additional discriminatory value between patients in this specific context, and we therefore did not include it in the baseline table.

  1. Additional clarifications

To further enhance the readability, adoptions have been made to figure 1 and figure 2, legends do now include descriptions.

Reviewer 2 Report

Comments and Suggestions for Authors

The authors presented results from a single low-volume center in Frozen Elephant Trunk (FET) and non-FET treatments for acute aortic dissection, conducted between October 2014 and December 2023. The results suggest that the FET procedure is practicable for low-volume centers equipped with a specialized aortic team. The work demonstrates considerable effort.

  1. In section 2.1, the authors stated, “Except in four cases, a 10cm stent graft length was used. Two patients received the E-vota open neo prosthesis in 18cm length and 2 patients the Thoraflex hybrid graft in 15cm length.” Please rephrase this to avoid potential misinterpretation.
  2. What p-value threshold did the authors choose? p<0.001 might be a bit confusing because some readers could interpret p = 0.001 as the threshold.
  3. Median (first quartile-third quartile) is not a common way to express the data, especially demographic data. Could the authors elaborate on why this is chosen instead of the mean ± standard deviation?
  4. Please add more descriptions to Figures 1 and 2 in their legends. Also, include the names and units of the x and y axes for improved clarity.
  5. Figure 3. ‘Survival rate’ instead of ‘survival’ might be a more accurate description. Please also include more information in the figure legend.

Author Response

Response to Reviewer 2 Comments

  1. Summary

1)         In section 2.1, the authors stated, “Except in four cases, a 10cm stent graft length was used. Two patients received the E-vota open neo prosthesis in 18cm length and 2 patients the Thoraflex hybrid graft in 15cm length.” Please rephrase this to avoid potential misinterpretation.

Thank you for your suggestions, we adopted this section to clarify. A 10 cm stentgraft was used in 62 patients, while 2 received an 18 cm E-vita open neo prosthesis and another 2 a 15 cm Thoraflex hybrid graft for anatomical reasons.

2)         What p-value threshold did the authors choose? p<0.001 might be a bit confusing because some readers could interpret p = 0.001 as the threshold.

We thank the reviewer for this comment. In our study, we defined statistical significance as p < 0.05. When p-values were smaller than 0.001, we reported them as p < 0.001 to indicate high significance, but the predefined threshold for significance throughout the analysis was p < 0.05. We have clarified this in the Methods section to avoid ambiguity.

3)         Median (first quartile-third quartile) is not a common way to express the data, especially demographic data. Could the authors elaborate on why this is chosen instead of the mean ± standard deviation?

We thank the reviewer for this important comment. We chose to present continuous variables as median (interquartile range) because many of the variables in our cohort were not normally distributed, as assessed by Shapiro–Wilk test. In this context, the median and interquartile range provide a more appropriate and robust measure of central tendency and spread than the mean ± standard deviation, which assumes normality.

4)         Please add more descriptions to Figures 1 and 2 in their legends. Also, include the names and units of the x and y axes for improved clarity.

We have adopted the legend of figure 1 and figure 2 according to your suggestions.

5)         Figure 3. ‘Survival rate’ instead of ‘survival’ might be a more accurate description. Please also include more information in the figure legend.

Thank you for your comment. The legend has been adopted.

  1. Additional clarifications

To further enhance the readability, adoptions have been made to figure 1 and figure 2, legends do now include descriptions.

Reviewer 3 Report

Comments and Suggestions for Authors

Despite the encouraging data presented by Centers with significant amount of cases, acute type A aortic dissection remains a challenge, often with a difficult outcome. Athough you present encouraging data, as your discussions suggest, a complex technique like FET, cannot be implemented easily in low volume centers, without taking into consideration the learning curve for the whole Heart team. I appreciate your results, but I do not see a very significant grade of novelty.

Author Response

Response to Reviewer 3 Comments

  1. Summary

Despite the encouraging data presented by Centers with significant amount of cases, acute type A aortic dissection remains a challenge, often with a difficult outcome. Athough you present encouraging data, as your discussions suggest, a complex technique like FET, cannot be implemented easily in low volume centers, without taking into consideration the learning curve for the whole Heart team. I appreciate your results, but I do not see a very significant grade of novelty.

We thank the reviewer for their comments. Acute type A aortic dissection remains challenging, especially in low-volume centers, and complex techniques like FET require careful team training. In our series, procedures were initially performed by three senior surgeons and gradually adopted to younger colleagues, ensuring a controlled learning curve. With proximalization of the FET and liberal use of extra-anatomical LSA bypass grafts in the beginning of the learning curve, we achieved outcomes comparable to high-volume centers. Despite longer HCA and SCP times, there was no negative impact on neurological outcomes or mortality compared to less extensive techniques, and completion TEVAR during follow-up was straightforward. While FET is not a novel concept, our study demonstrates its safe and effective adoption in low-volume centers through technical adaptations.

Reviewer 4 Report

Comments and Suggestions for Authors

Voetsch et al. present a retrospective cohort study examining the implementation of the FET technique for acute aortic dissection in a low-volume cardiac surgery center. While the manuscript addresses an important clinical question—whether complex aortic procedures can be safely performed outside high-volume specialist centers—it is limited by significant methodological and interpretive shortcomings.

First, the statistical analysis requires substantial refinement. The reported 10% absolute mortality difference (13% vs 23%, p=0.068) translates into a 77% relative increase in mortality risk, which is clinically meaningful despite not achieving conventional statistical significance. The authors’ conclusion of “comparable outcomes” based solely on p>0.05 reflects a misunderstanding of the distinction between statistical significance and clinical relevance.

Second, the assertion that “follow-up was available in all patients” over a 9-year period in 202 cases is methodologically implausible without detailed verification procedures. Complete follow-up is exceedingly rare, even in prospective studies with dedicated infrastructure, and typically requires systematic tracking protocols (e.g., national registries, repeated contact attempts, family interviews). The absence of reported losses to follow-up or missing data undermines the study’s credibility.

Third, the marked evolution of surgical practice over the study period—including a decline in proximalization from >80% to <40% and in LSA bypass from >60% to <20%—introduces temporal confounding that fundamentally challenges the validity of aggregated outcome comparisons.

The manuscript also contains grammatical errors (example, in abstract “lerning curve” instead of “learning curve”) that require correction.

More importantly, while the authors focus on institutional case volume, it should be acknowledged that in technically demanding procedures such as FET, surgical expertise may outweigh volume effects. Notably, the inclusion of complex concomitant procedures such as valve-sparing aortic root replacement (David procedure) in a low-volume setting raises concerns, given the likely limited surgical experience. This further undermines the strength of the authors’ conclusions.

Finally, I recommend that the discussion be expanded to incorporate recent literature on FET outcomes, including:

Journal of Clin. Med. 2025, 14(14), 5170; https://doi.org/10.3390/jcm14145170

In its current form, the manuscript addresses a clinically relevant topic but requires significant methodological clarification, nuanced interpretation of mortality data, acknowledgment of surgical expertise as a confounding factor, and integration of more recent evidence.

Comments on the Quality of English Language

NA

Author Response

Response to Reviewer 4 Comments

  1. Summary

First, the statistical analysis requires substantial refinement. The reported 10% absolute mortality difference (13% vs 23%, p=0.068) translates into a 77% relative increase in mortality risk, which is clinically meaningful despite not achieving conventional statistical significance. The authors’ conclusion of “comparable outcomes” based solely on p>0.05 reflects a misunderstanding of the distinction between statistical significance and clinical relevance.

We thank the reviewer for emphasizing this important distinction. We fully acknowledge that statistical non-significance (p>0.05) should not be interpreted as clinical equivalence, and we do not imply that FET procedures are inherently equivalent or comparable regarding operative risk to less extensive aortic repairs. Our conclusion of “comparable outcomes” is intended to reflect the feasibility and safety of FET implementation in a low-volume center, based on perioperative neurological outcomes, procedural success, and alignment with predicted operative risk scores compared to current literature. Furthermore, patients undergoing extensive FET repair for radiographic present malperfusion represent a higher-risk cohort than those undergoing more limited dissections, precluding direct equivalence comparisons between groups.

We expanded the limitations section to clarify this study was not designed and powered to compare outcomes between FET and non-FET patients. “It should be emphasized that this study was not designed or powered to compare outcomes between FET and less extensive aortic repair groups, and any observed differences should be interpreted in the context of differing baseline risk profiles and procedural complexity. Both groups were well balanced in their baseline characteristics except signs of malperfusion and arch entry tears which account for most FET indications.”

Second, the assertion that “follow-up was available in all patients” over a 9-year period in 202 cases is methodologically implausible without detailed verification procedures. Complete follow-up is exceedingly rare, even in prospective studies with dedicated infrastructure, and typically requires systematic tracking protocols (e.g., national registries, repeated contact attempts, family interviews). The absence of reported losses to follow-up or missing data undermines the study’s credibility.

We thank the reviewer for raising this point. Follow-up in our cohort was systematically obtained through a dedicated aortic outpatient clinic and supplemented by telephone interviews, allowing for comprehensive tracking of all 202 patients over the 9-year period. While we acknowledge that complete follow-up over such a timeframe is challenging, our structured approach ensured no patients were lost to follow-up, and all relevant outcomes were recorded. We will clarify these procedures in the Methods section to improve transparency and address the reviewer’s concern.

Third, the marked evolution of surgical practice over the study period—including a decline in proximalization from >80% to <40% and in LSA bypass from >60% to <20%—introduces temporal confounding that fundamentally challenges the validity of aggregated outcome comparisons.

We thank the reviewer for this observation. As with all complex surgical therapies, technical adaptations are often implemented progressively to facilitate adoption and ease the learning curve. In our series, proximalization of the FET and liberal use of LSA bypass grafts were initially applied to simplify the procedure, consistent with recommendations from other groups. With growing team experience and familiarity with extensive aortic repair, these adaptations became less necessary, reflecting a natural evolution of surgical practice rather than a limitation in study validity.

The manuscript also contains grammatical errors (example, in abstract “lerning curve” instead of “learning curve”) that require correction.

Thank you for carefully reading the manuscript. We checked for grammatical errors.

More importantly, while the authors focus on institutional case volume, it should be acknowledged that in technically demanding procedures such as FET, surgical expertise may outweigh volume effects. Notably, the inclusion of complex concomitant procedures such as valve-sparing aortic root replacement (David procedure) in a low-volume setting raises concerns, given the likely limited surgical experience. This further undermines the strength of the authors’ conclusions.

We thank the reviewer for this comment. Literature suggests that an annual caseload of 30–40 AAD cases provides adequate outcomes, highlighting the relationship between volume and surgical results (Kawczynski MJ et al., Eur J Cardiothorac Surg, 2025;67:ezaf022). However, only a few surgeons in Europe perform this number of acute type A aortic dissection (AAD) cases annually, with most centers managing well below 100 cases per year. In the United States, in 2014, 44% of hospitals performed fewer than 20 open AAD surgeries annually (https://doi.org/10.1161/JAHA.118.011402).

We fully agree that surgical expertise may outweigh volume effects in complex procedures. In our center, despite a relatively low annual volume of 20–30 acute dissections, the senior surgeons had extensive prior experience in aortic root and arch surgery, including a substantial number of AAD cases, before initiating FET in acute dissections. Through gradual adoption of technical adaptations, such as proximalization of the FET and selective LSA bypass, this expertise enabled the safe and effective implementation of FET procedures in a low-volume setting.

Finally, I recommend that the discussion be expanded to incorporate recent literature on FET outcomes, including:

Journal of Clin. Med. 2025, 14(14), 5170; https://doi.org/10.3390/jcm14145170

We adopted the discussion and included the suggested meta-analysis.

Round 2

Reviewer 4 Report

Comments and Suggestions for Authors

The comments I raised in the first round have been adequately addressed. I particularly appreciate the limitation added by the authors as well as the expansion of the Discussion section.